# Predictions of Moisture Diffusion Behavior of Cellulose-Fiber-Reinforced Plain Weave Epoxy Composites

**DOI:** 10.3390/polym13142347

**Published:** 2021-07-17

**Authors:** Hang Yu, Jie Yang

**Affiliations:** School of Transportation and Civil Engineering, Nantong University, Nantong 226019, China; yuhang@ntu.edu.cn

**Keywords:** superposition method, jute fiber, moisture uptake, plain weave composite, three-phase diffusion model

## Abstract

Cellulose-fiber-reinforced plain weave composites absorb lots of water from humid environments because of their inherent susceptibility to moisture. Moisture absorption experiments with cellulose fiber plain weave composites have been reported by some researchers; however, few theoretical studies have been performed to date to predict their moisture diffusion behavior. In this paper, the moisture diffusion behavior of cellulose-fiber-reinforced plain weave composite is predicted using a novel superposition method considering its microweave pattern. The overall moisture uptake of the composite is treated as moisture absorption superposition of the fiber bundles part, resin part, undulated fiber bundles and resin-rich part in the unit cell. The moisture diffusion of the undulated fiber bundles and resin-rich part is more complicated than the other parts; thus, a solution for a unique three-phase diffusion problem is used to solve this special moisture diffusion issue. Both finite element analysis and experiments are carried out to validate the proposed approach, with the results showing that the predictions can effectively characterize the moisture diffusion behavior of cellulose-fiber-reinforced plain weave composites.

## 1. Introduction

Due to increasing pressure related to environmental concerns, more recycled materials are required. Cellulose fibers (such as curaua, sisal and jute) offer the advantages of low relative density, low cost and good recyclability compared with synthetic fibers [1,2,3], so their composites hold promise for future use in automotive, civil engineering and other fields [4,5,6,7,8,9]. Despite these advantages, the main drawback of cellulose fiber composites is their high hydrophilicity when they are applied in humid environments [10,11]. Some studies on the moisture diffusion behavior of these composites have been performed, investigating the fiber–matrix properties [12,13], surface treatment approaches [14,15] and fiber configurations [16,17], which play significant roles in their moisture diffusion behaviors. Additionally, the weave patterns of plain weave or woven composites also make important contributions to their moisture diffusion properties [18]. The undulated fiber bundles and resin-rich regions in these composites make the moisture permeation process complicated [19]. For high-performance applications, the moisture diffusion behaviors of these composites should be effectively predicted to better support their long-term service [20,21].

The moisture diffusion behaviors of unidirectional cellulose fibers and synthetic fiber composites almost obey Fick’s law [22,23,24]. Li and Xue [22] studied the moisture diffusion mechanism of unidirectional jute-fiber reinforced-composites and found that the water absorption follows Fickian behavior, while the unique microstructure and composition effects on the degradation of the jute fibers were also discussed. Saidane et al. [23] investigated the moisture diffusion behavior of unidirectional jute-fiber-reinforced composites, concluding that the moisture diffusion in each direction of the composite follows Fickian behavior, while the water absorption in the fiber direction is higher than in other directions because the hollow part in its morphology offers an easier path for moisture diffusion. Almeida et al. [24] studied the moisture diffusion behavior of carbon-fiber-reinforced, epoxy-filament-wound composite laminates exposed to hydrothermal conditioning, with the results showing that Fickian diffusion occurs for these composites.

Non-Fickian behaviors were found in both cellulose fiber and synthetic fiber woven composites [18,19,25,26,27]. Perrier et al. [25] investigated the water uptake of hemp-fiber-reinforced plain weave composites, showing that the weave pattern had an influence on moisture diffusion, while non-Fickian absorption was observed, which have may been related to the different types of fiber reinforcement. Cherif et al. [18] found that the classical Fickian model cannot correctly describe water diffusion in flax-fiber-reinforced woven composites. An initial fast diffusion stage followed by a slow diffusion stage were observed in two kinds of specimens with different weave patterns. The two-phase diffusion model was proposed to represent two independent cases of Fick’s diffusion; the first phase is related to quick water diffusion due to fiber and interface porosity, while the second phase corresponds to unopened macroporosities in the materials. Bao and Yee [19] discussed the non-Fickian behavior in plain weave composites of synthetic fibers. The observed quick diffusion may have been associated with voids, while the slow stage was related to the moisture diffusion of the resin phase due to resin-rich pockets. A dual-diffusivity model was then successfully established to describe this non-Fickian behavior. Eggers et al. [27] investigated the moisture diffusion behavior of filament-wound composite rings under harsh environments. The initial experiment results at room temperature and with hot water fit Fick’s law, while ±60/±90 composites showed non-Fickian behavior, namely fast transient diffusion at the beginning followed by a slower diffusion rate.

Although some analytical methods, such as the Langmuir model and dual-diffusivity model, can predict the moisture diffusion behavior of both cellulose fiber and synthetic fiber woven composites, few theoretical models have been developed considering the effects of undulated fiber bundles and resin pockets in the microweave structure of these composites. In this paper, the moisture diffusion behavior of cellulose fiber plain weave composites was investigated theoretically using a superposition method, and the water absorption levels of the undulated fiber tow and resin-rich parts were also modeled using a three-phase diffusion model. Finite element analysis and moisture uptake experiments were carried out on the jute-fiber-reinforced composites to validate the proposed model.

## 2. Method and Experiment

### 2.1. Theoretical Method

#### 2.1.1. Geometry of Plain Weave Composite

Figure 1 illustrates a classic plain weave geometry, whereby a crossed weft and warp tows compose the reinforced phase of the composite. Due to the periodicity in the microstructure in the composites, the total moisture uptake of the composites can be represented by the unit cell depicted in Figure 1a. The unit cell strategy is an effective method for dealing with different mechanical property issues in plain weave composites.

For moisture diffusion problems involving plain weave composites through the thickness direction, the complicated geometry of the unit cells make the theoretical prediction models hard to develop; therefore, the cross-sections of the weft and warp tows in this paper are treated as rectangular and the undulated fiber tows are assumed to be cubes to simplify the analysis. The simplified unit cells composing the plain weave composite and the geometry parameters are shown in Figure 1b. Here, *a*, *b*, *c* and *θ* are the represented tow thickness, tow width, tow distance and tow weave angle, respectively.

#### 2.1.2. Superposition Method for Moisture Diffusion of Plain Weave Composite

Since the moisture diffusion in different regions in the plain composite through the thickness direction is microstructure-dependent, analysis of the water immersion is achieved using the superposition method for each independent part. Here, we assume that the moisture exchange between each adjacent part can be neglected, as the total moisture uptake of the unit cell is a superposition of each part.

The superposition method is depicted in Figure 2. The orange regions are fiber tows, while the empty regions in the unit cell are resin materials. This unit cell is divided into 9 parts according to the type of each structure, the details of which are shown in Figure 2a. Parts 1, 3, 7 and 9 in Figure 2b represent tow parts, which can be treated as unidirectional composites. The moisture in these parts actually diffuses through the transverse direction of unidirectional composites. As shown in Figure 2c, we found that parts 2, 4, 6 and 8 had a similar structure, i.e., the upper and lower faces in these parts were resin materials, while the middle structures in these parts were undulated tows. If *c* = *d*, we would find the moisture absorption of these parts to be equal. The transient moisture diffusion in parts 2, 4, 6 and 8 may be more complicated than in other parts, due to the undulated tows and different types of materials in these parts; thus, the prediction of moisture diffusion is a key problem in this paper. Part 5 is a pure resin material, which is shown in Figure 2d. Its moisture uptake obeys Fick’s law.

For moisture diffusion through the thickness direction in part 4, the analysis of the simplified equivalence structure of part 4 is illustrated in Figure 3. We first cut part 4 (Figure 3a) in half due to its symmetrical properties, whereby half of part 4 is depicted in Figure 3b. Obviously, the moisture absorption of part 4 is twice of half of part 4. From Figure 3b, it can be seen that water immersion in upper face of the half-part will first diffuse through resin regions; however, water will also diffuse into the orange region at the beginning. This phenomenon will make the water absorption in the initial stage complicated. If the orange region represents the cellulose fiber tow, the moisture diffusion of the half-part will increase slowly at the initial stage, since the diffusivity of the cellulose fiber tow is larger than the resin phase. Then, the moisture diffusion will quickly increase because of the cellulose fiber tow. To predict this moisture diffusion behavior, the fast diffusion of cellulose fiber tow and slow diffusion of resin material are considered here by using a simplified equivalence method. The undulated fiber tow and triangular and trapezoidal resin phases are assumed to be equivalent to rectangle structures, so that the analysis is simplified as a three-phase diffusion problem. The areas of different phases in the equivalent model are equal to the former fiber tow and resin regions. Next, we will talk about three thickness cases for the upper resin.

Minimum equivalence. If we directly ensure that the area of the upper triangular resin phase is equal to the equivalent model, i.e., the thickness of the upper resin in the equivalent model is 4/a, the moisture diffusion will be underestimated because moisture diffusion in the cellulose fiber tow is not enough to be considered. We will later prove this phenomenon; this model is named the minimum equivalence model, as shown in Figure 3c.Maximum equivalence. If the thickness of the upper resin phase is 0, as depicted in Figure 3d, the moisture diffusion will be obviously overestimated. This model is named the maximum equivalence model.Approximate equivalence. As illustrated in Figure 3e, the moisture uptake of half of part 4 is actually situated between minimum equivalence and minimum equivalence. The thickness of the upper resin phase is *τa.* This model is named the approximate equivalence model.

The total moisture uptake of the unit cell is the water absorption superposition of part 1 *M*_P1_(*t*), part 4 *M*_P4_(*t*) and part 5 *M*_P5_(*t*). Since *M*_P4_(*t*) is equal to twice the moisture uptake of approximate equivalence of half of part 4 *M*_Pe4_(*t*), the total moisture uptake of the unit cell is given as:(1)$$M\left(t\right)=4M_{P1}\left(t\right)+8M_{\mathrm{Pe}4}\left(t\right)+M_{P5}\left(t\right)$$

#### 2.1.3. Three-Phase Diffusion Equation

The three-phase diffusion model for part 4 is shown in Figure 4. The saturated moisture concentration and diffusivity of the upper and bottom phase are *C*_1_ and *D*_1_, respectively. The saturated moisture concentration and diffusivity of the middle phase are *C*_2_ and *D*_2_, respectively. The thicknesses of phases 1, 2 and 3 are *a_1_-a_2_, a_2_-a_3_* and *a_3_*, respectively. The moisture concentrations in phases 1, 2 and 3 are *C*_1_*(x,t)*, *C*_2_*(x,t)* and *C*_3_*(x,t)*, respectively, and their moisture diffusion equations are:
(2)$$\frac{\partial C_{1}(x,t)}{\partial t}=D_{1}\frac{\partial^{2}C_{1}(x,t)}{\partial x^{2}},\;a_{2}<x<a_{1},\;t>0$$
(3)$$\frac{\partial C_{2}(x,t)}{\partial t}=D_{2}\frac{\partial^{2}C_{2}(x,t)}{\partial x^{2}},\;a_{3}<x<a_{2},\;t>0$$
(4)$$\frac{\partial C_{3}(x,t)}{\partial t}=D_{1}\frac{\partial^{2}C_{3}(x,t)}{\partial x^{2}},\;0<x<a_{3},\;t>0$$

The boundary conditions are:(5)$$C_{1}(a_{1},t)=C_{1},t\geq 0$$
(6)$$C_{3}(0,t)=C_{1},t\geq 0$$

The initial conditions are:(7)$$C_{1}(x,0)=0,\;a_{2}<x<a_{1}$$
(8)$$C_{2}(x,0)=0,\;a_{3}<x<a_{2}$$
(9)$$C_{3}(x,0)=0,\;0<x<a_{3}$$

The interface conditions are:(10)$$\begin{array}{l} \frac{C_{1}(a_{2},t)}{C_{1}}=\frac{C_{2}(a_{2},t)}{C_{2}},\;t\geq 0\\ D_{1}\frac{\partial C_{1}(a_{2},t)}{\partial x}=D_{2}\frac{\partial C_{2}(a_{2},t)}{\partial x},\;t\geq 0 \end{array}$$
(11)$$\begin{array}{l} \frac{C_{2}(a_{3},t)}{C_{2}}=\frac{C_{3}(a_{3},t)}{C_{1}},\;t\geq 0\\ D_{2}\frac{\partial C_{2}(a_{3},t)}{\partial x}=D_{1}\frac{\partial C_{3}(a_{3},t)}{\partial x},\;t\geq 0 \end{array}$$

The solutions are:(12)$$C_{1}(x,t)=C_{1}\left(1+\sum_{m=1}^{\infty }\frac{2d_{1}}{\beta_{m}d_{4}}e^{-D_{1}\beta_{m}^{2}t}\right)$$
(13)$$C_{2}(x,t)=C_{2}\left(1+\sum_{m=1}^{\infty }\frac{4d_{2}}{\beta_{m}d_{4}}e^{-D_{1}\beta_{m}^{2}t}\right)$$
(14)$$C_{3}(x,t)=C_{1}\left(1+\sum_{m=1}^{\infty }\frac{2d_{3}}{\beta_{m}d_{4}}e^{-D_{1}\beta_{m}^{2}t}\right)$$
where $d_{1}$, $d_{2}$, $d_{3}$ and $d_{4}$ are given in Appendix A. Here, we define $\mu =i\beta_{m}$ (m=1,2,3,…) as the root of the equation below; thus, $\beta_{m}$ can be obtained by following the below equation:(15)$$\begin{array}{l} \left(1-\sigma^{2}r^{2}\right)\sin\beta_{m}\left[-\left(a_{1}-a_{2}-a_{3}\right)+k\left(a_{2}-a_{3}\right)\right]+\left(1+2\sigma r+\sigma^{2}r^{2}\right)\sin\beta_{m}\left[\left(a_{1}-a_{2}+a_{3}\right)+k\left(a_{2}-a_{3}\right)\right]+\\ \left(1-2\sigma r+\sigma^{2}r^{2}\right)\sin\beta_{m}\left[-\left(a_{1}-a_{2}+a_{3}\right)+k\left(a_{2}-a_{3}\right)\right]+\left(1-\sigma^{2}r^{2}\right)\sin\beta_{m}\left[\left(a_{1}-a_{2}-a_{3}\right)+k\left(a_{2}-a_{3}\right)\right]=0 \end{array}$$
where $\sigma =\left(kD_{2}/D_{1}\right)$, $k={\left(D_{1}/D_{2}\right)}^{1/2}$,$C_{2}/C_{1}=r$. Then, the moisture absorption levels of the three phases are:(16)$$M_{1}(t)=\int_{a_{2}}^{a_{1}}C_{1}(x,t)dx=C_{1}\left(a_{1}-a_{2}+\sum_{m=1}^{\infty }\frac{2d_{5}}{\beta_{m}^{2}d_{4}}e^{-D_{1}\beta_{m}^{2}t}\right)$$
(17)$$M_{2}(t)=\int_{a_{3}}^{a_{2}}C_{2}(x,t)dx=C_{2}\left(a_{2}-a_{3}+\sum_{m=1}^{\infty }\frac{4d_{6}}{k\beta_{m}^{2}d_{4}}e^{-D_{1}\beta_{m}^{2}t}\right)$$
(18)$$M_{3}(t)=\int_{0}^{a_{3}}C_{3}(x,t)dx=C_{1}\left(a_{3}+\sum_{m=1}^{\infty }\frac{2d_{7}}{\beta_{m}^{2}d_{4}}e^{-D_{1}\beta_{m}^{2}t}\right)$$
where, $d_{5}$, $d_{6}$ and $d_{7}$ are given in Appendix A.

The total moisture uptake $M_{\mathrm{Pe}4}\left(t\right)=M_{1}\left(t\right)+M_{2}\left(t\right)+M_{3}\left(t\right)$.

### 2.2. Experiment

#### 2.2.1. Experimental Material and Setup

Jute fibers have good specific mechanical properties and are widely used in many fields; thus, jute fibers were selected as the reinforcement in plain weave composites in this paper. The jute fiber fabrics were provided by Nanjing Hitech Composites Co., Ltd. (Nanjing, China) The compression molding approach was used to manufacture composites. The epoxy-impregnated plain weave jute fiber composites were placed in the mold to cure at room temperature for 48 h. Two types of jute fiber plain weave composites were made, and the composite plates were cut and shaped to rectangular form using a diamond saw blade (Nanjing Hitech Composites Co., Ltd, Nanjing, China). The geometry parameters of the specimens are given in Table 1. The water absorption experiment was performed after the initial drying.

#### 2.2.2. Water Absorption Experiment

The water absorption experiment was implemented after the specimens were dried in an oven (Nanjing Hitech Composites Co., Ltd, Nanjing, China) at 60 °C for 24 h. The specimens were taken into an environmental chamber (Nanjing Hitech Composites Co., Ltd, Nanjing, China) with a relative humidity (RH) of 100% at 60 °C. The waterproof coating was used to clog the lateral surfaces of specimens to ensure the moisture diffusion through the thickness direction. The specimens were periodically taken out of the chamber during the aging process. To evaluate the weight increase, the specimens were wiped dry with tissue paper and weighed using an electronic balance. The weighing was repeated up to saturation, which meant that the specimens did not show significant variations in mass. The water absorption of the composite was evaluated using the relative uptake of the weight according to:(19)$$M_{t}=\frac{W_{t}-W_{0}}{W_{0}}\times 100\%$$
where $W_{0}$ and $W_{t}$ are the weight of the dry and wet specimens at time t. To calculate the diffusion parameters for one dimension, Fick’s diffusion was needed, the expression of which expression was given as follows:(20)$$M_{t}=M_{m}\left\{1-\sum_{n=0}^{\infty }\frac{8}{\pi^{2}{\left(2n+1\right)}^{2}}\exp\left[-D{\left(\frac{2n+1}{h}\pi \right)}^{2}t\right]\right\}$$
where *h* is the specimen thickness, *D* is the diffusivity and *M_m_* is its maximum moisture uptake in equilibrium state. The diffusivity *D* in Fick’s law is given by:(21)$$D=\pi {\left(\frac{hk}{4M_{m}}\right)}^{2}$$
where k is the slope of the linear part of $M_{t}$ versus the $\sqrt{t}$ curve.

## 3. Results and Discussion

### 3.1. Validation of Three-Phase Diffusion Model

The three-phase diffusion equation was solved using Equations (16)–(18). We first utilized FEA to validate the exact solution. Two cases for the moisture uptake for part 4 were considered: one case was *D*_1_ (0.005) < *D*_2_ (0.02), *C*_1_ (0.5) < *C*_2_ (1); the other case was *D*_1_ (0.02) > *D*_2_ (0.005), *C*_1_ (1) > *C*_2_ (0.5). Two different cases were validated using FEA with different phase thicknesses, the results of which are shown in Figure 5 and Figure 6 and the dimension parameters of which are shown in Table 2.

It can be seen that the FEA results are in good agreement with the proposed three-phase model, with various scales for different phases. To obtain correct theoretical results, the boundary condition should use continuous normalized concentration using Equations (10) and (11). This boundary condition was discussed in our last article [28].

### 3.2. Comparison with FEA

#### 3.2.1. The Determination of Value τ

The value τ may be affected by the thickness, dimensions (c/a) and diffusivities of different materials in part 4. To obtain the approximate equivalent structure of part 4, the fiber material, which is cellulose fiber $D_{1}\left(0.005\right)<D_{2}\left(0.02\right)$, was firstly examined, the results of which are shown in Figure 7 and Figure 8. For different dimensions, we found that if τ=1/8, the three-phase model prediction coincided with FEA; τ=1/4 represents the minimum equivalence, the moisture uptake of which is smaller than the FEA data. The reason for this is that the equivalence underestimates the fast diffusion in cellulose fiber tows. If τ=1/16, the equivalence immerses the moisture into the structure too quickly, so that the theoretical calculations are faster than the FEA data.

If the fiber material is synthetic fiber, $D_{1}\left(0.02\right)>D_{2}\left(0.005\right)$. Comparisons between the theoretical and FEA results are shown in Figure 9 and Figure 10. From the results, it can be seen that the value τ=1/4, 1/8, 1/16 will have less effect on the moisture uptake with different dimensions of part 4 compared with the case ($D_{1}<D_{2}$). This is because the domain moisture contribution is in resin phase and slight changes of the phase thickness will have less effect on the moisture uptake; thus, we also let τ=1/8, which could be the upper face thickness condition for the approximate equivalence.

From the analysis above, the conclusion was draw that if τ=1/8, the theoretical model will agree with FEA data for both cellulose fiber and synthetic fiber cases. The approximate equivalence represents the right balance between the minimum equivalence and maximum equivalence, τ=1/8, which approximately simulated the fast diffusion and slow diffusion in part 4.

#### 3.2.2. Moisture Uptake of the Whole Unit Cell

The moisture diffusivities of undulated fiber tows are related to their longitudinal and transverse diffusivities. The longitudinal diffusion diffusivity *D*_1_ of the cellulose-fiber-reinforced composite was generally larger than the transverse diffusion diffusivity D_2_, since the hollow middle part of the cellulose fiber in the longitudinal direction, called the lumen, provided a fast path for moisture immersion [23]. The moisture diffusion diffusivity of declining fiber tows in part 4 can be calculated by:(22)$$D_{z}=D_{1}\sin^{2}\theta_{p}+D_{2}\cos^{2}\theta_{p}$$

Using the geometry parameters of each material in the unit cell as shown in Table 3, the moisture uptake of the whole unit cell was obtained by Equation (1) and the FEA results were compared with theoretical calculations, as depicted in Figure 11. From the comparison, we could find that the predictions using the proposed model fit well with the FEA results, proving that the superposition method could be an effective way to predict the moisture uptake of plain weave composites.

### 3.3. Experimental Validation

Jute fibers have good specific mechanical properties and are widely used in many fields; thus, jute fibers were selected as the reinforcement for plain weave composites in this paper. The moisture diffusion parameters of the specimens are given in Table 3 and Table 4. In Table 4, the transverse diffusivity *D*_2_ and saturated moisture content *M*_f_ of the jute fiber composite are about 4 and 6 times larger than resin materials *D*_r_ and *M*_r_, respectively, resulting in fast moisture diffusion in part 1, 3, 7 and 9 and slow moisture diffusion in the upper resin phase in parts 2, 4, 5, 6 and 8.

The moisture diffusion behaviors of composite a and b were tested in the experiment and comparisons with the theoretical results are given in Figure 12 and Figure 13. It can be seen that the proposed method matches the experimental results for both plain weave composites, with slow moisture diffusion being found corresponding to Fick’s law. The initial fast moisture diffusion was associated with the horizontal jute fiber tows, which had higher diffusion diffusivity. Although the middle undulated fiber tows had higher diffusivity, the following slow moisture diffusion was caused by resin-rich areas, which decreased the overall moisture diffusion in the composites.

## 4. Conclusions

In this paper, the moisture diffusion behavior of jute-fiber-reinforced plain weave composites was predicted using a superposition method and considering the microstructures in the composites. The experimental data showed good agreement with the theoretical model, proving that the proposed model effectively described the moisture diffusion behaviors of jute-fiber-reinforced plain weave composites. The moisture diffusion in jute fiber bundles was obviously faster than that of resin. It was found that the slow moisture diffusion in the upper and bottom face resin regions caused a decrease in the moisture uptake tendency.

The three-phase diffusion equation was developed to model moisture diffusion in resin-rich and undulated fiber tows regions. This equation demonstrated that the theoretical calculations fit the FEA results well. The three-phase diffusion equation may also have potential applications in moisture absorption problems in sandwich structures with different phase thicknesses and in undulated fiber tow structures in woven composites.

## Figures and Tables

**Figure 1 polymers-13-02347-f001:**
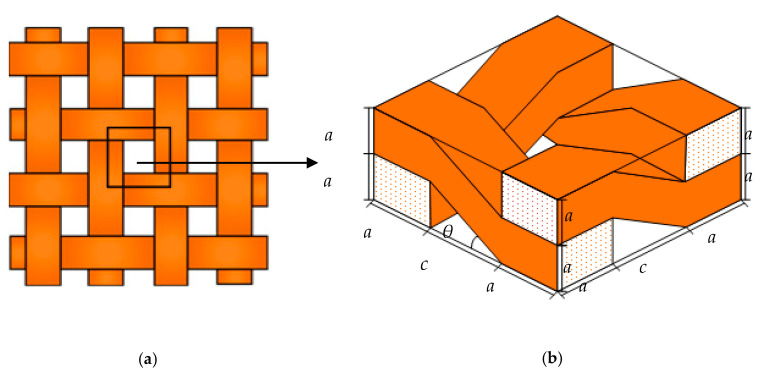
Idealized unit cell of a plain weave composite and its geometry parameters. (**a**) Unit cell of a plain weave composite; (**b**) Geometry parameters of the unit cell.

**Figure 2 polymers-13-02347-f002:**
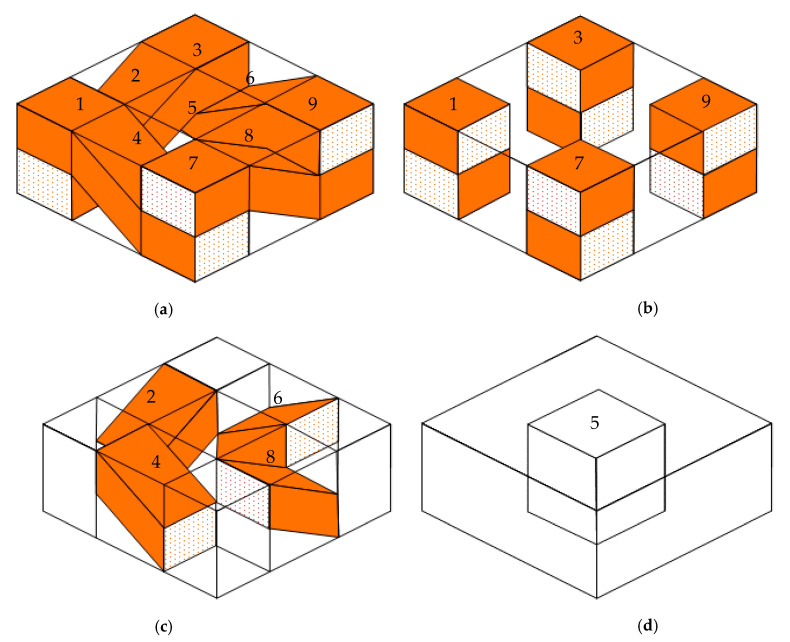
Superposition method for plain weave composite. (**a**) Partition of unit cell; (**b**) Tow part; (**c**) Undulated tow and resin part; (**d**) Resin part.

**Figure 3 polymers-13-02347-f003:**
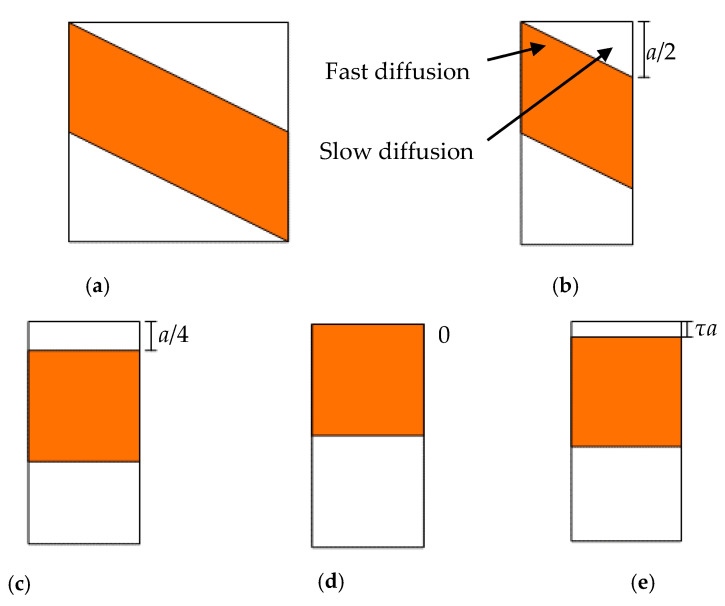
Moisture diffusion and equivalence structure analysis of part 4. (**a**) Part 4; (**b**) 1/2 Part 4; (**c**) Minimum equivalence; (**d**) Maximum equivalence; (**e**) Approximate equivalence.

**Figure 4 polymers-13-02347-f004:**
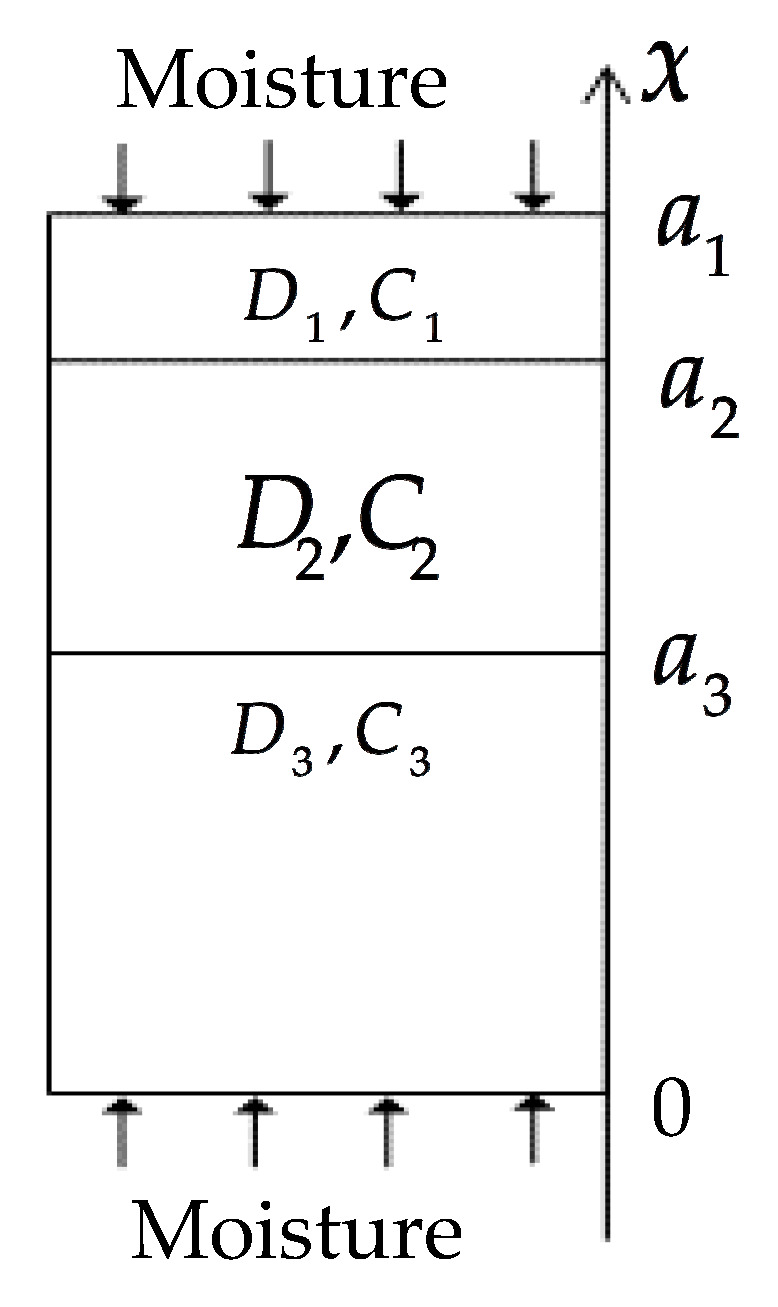
Three-phase diffusion model.

**Figure 5 polymers-13-02347-f005:**
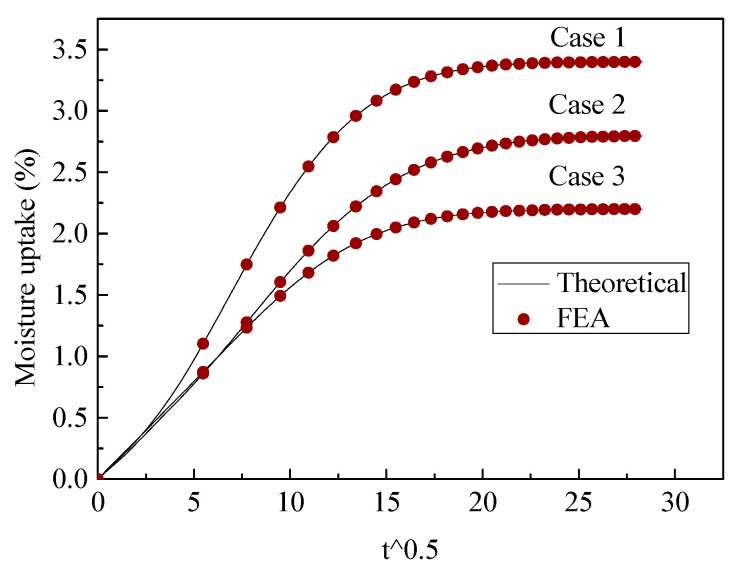
Validation of three-phase diffusion model, $D_{1}\left(0.005\right)<D_{2}\left(0.02\right)$, $C_{1}\left(0.5\right)<C_{2}\left(1\right)$.

**Figure 6 polymers-13-02347-f006:**
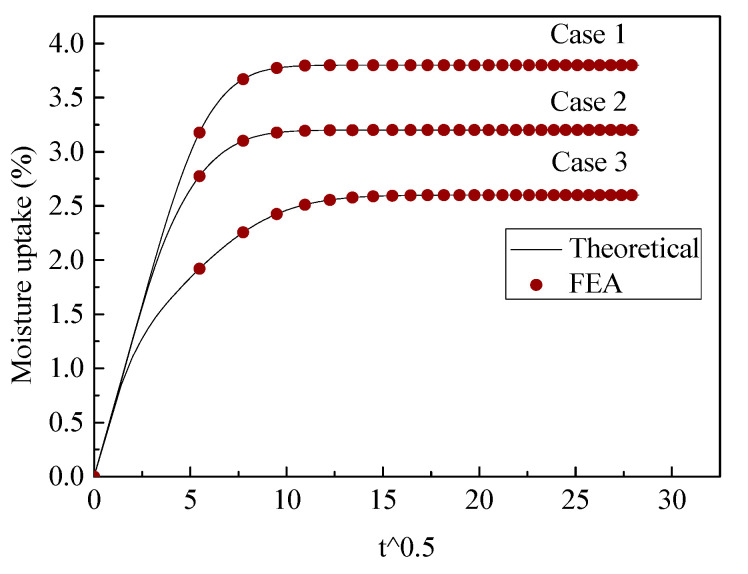
Validation of three-phase diffusion model, $D_{1}\left(0.02\right)>D_{2}\left(0.005\right)$, $C_{1}\left(1\right)>C_{2}\left(0.5\right)$.

**Figure 7 polymers-13-02347-f007:**
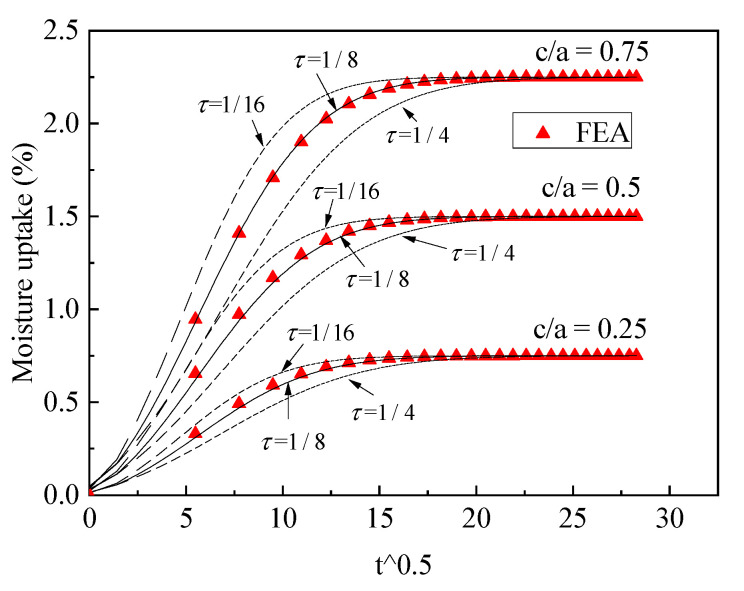
Validation of the three-phase diffusion model, $D_{1}<D_{2}$, c/a<1.

**Figure 8 polymers-13-02347-f008:**
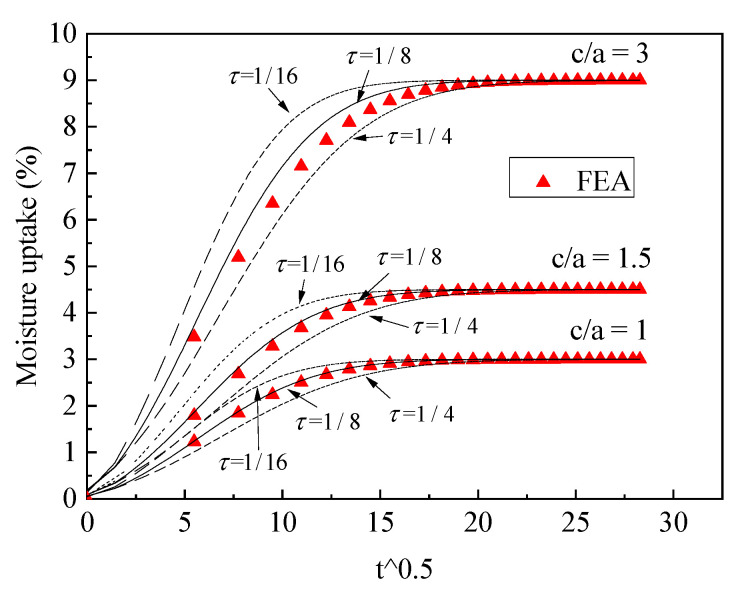
Validation of the three-phase diffusion model, $D_{1}<D_{2}$, c/a≥1.

**Figure 9 polymers-13-02347-f009:**
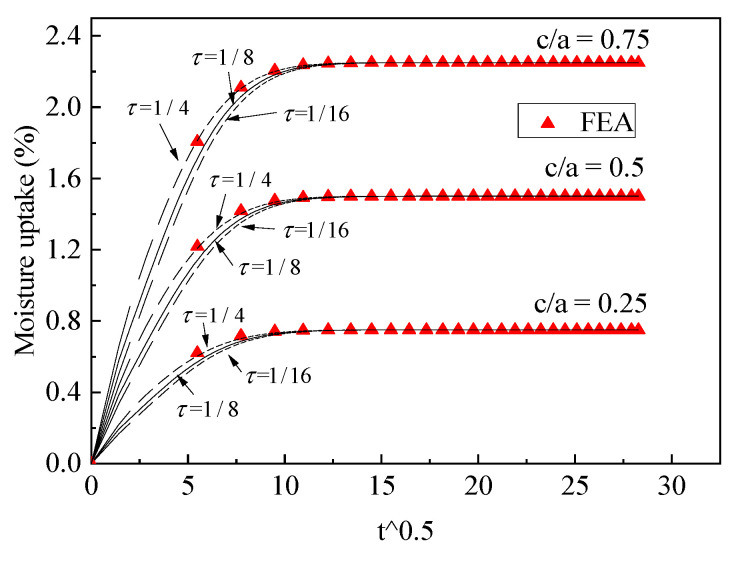
Validation of the three-phase diffusion model, $D_{1}>D_{2}$, c/a<1.

**Figure 10 polymers-13-02347-f010:**
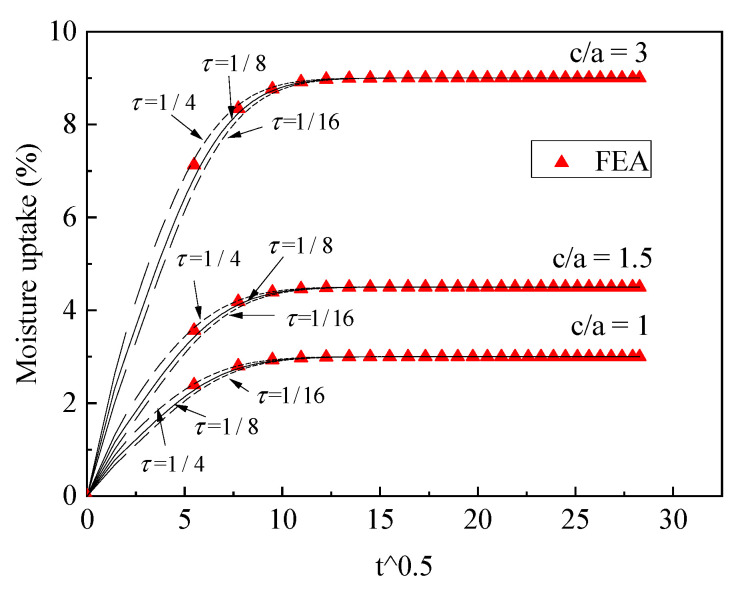
Validation of the three-phase diffusion model, $D_{1}>D_{2}$, c/a≥1.

**Figure 11 polymers-13-02347-f011:**
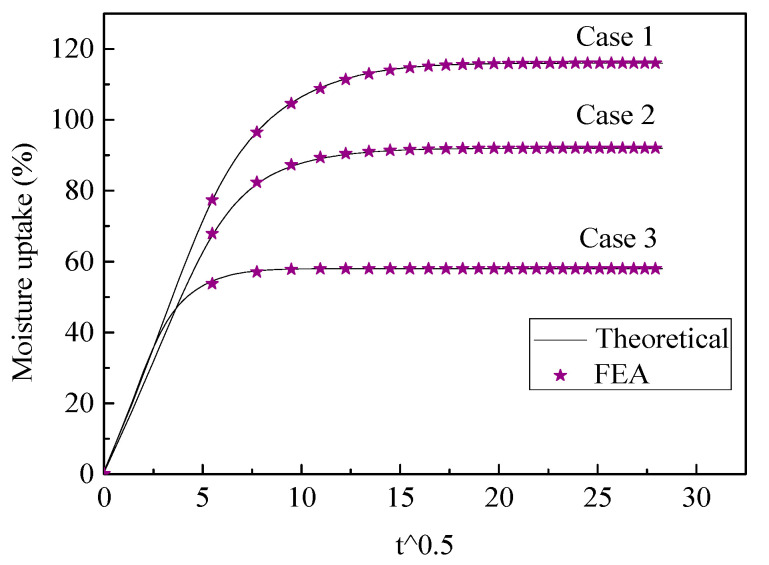
FEA and theoretical calculation of the whole unit cell.

**Figure 12 polymers-13-02347-f012:**
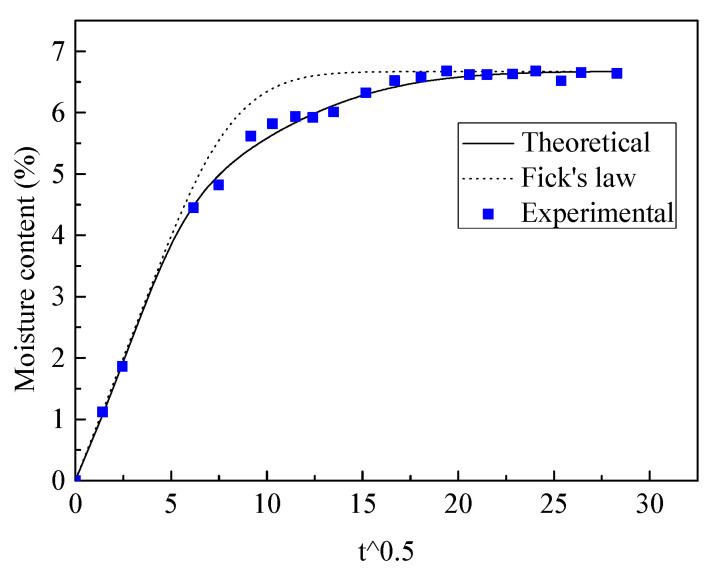
The experimental and theoretical moisture diffusion results for composite a.

**Figure 13 polymers-13-02347-f013:**
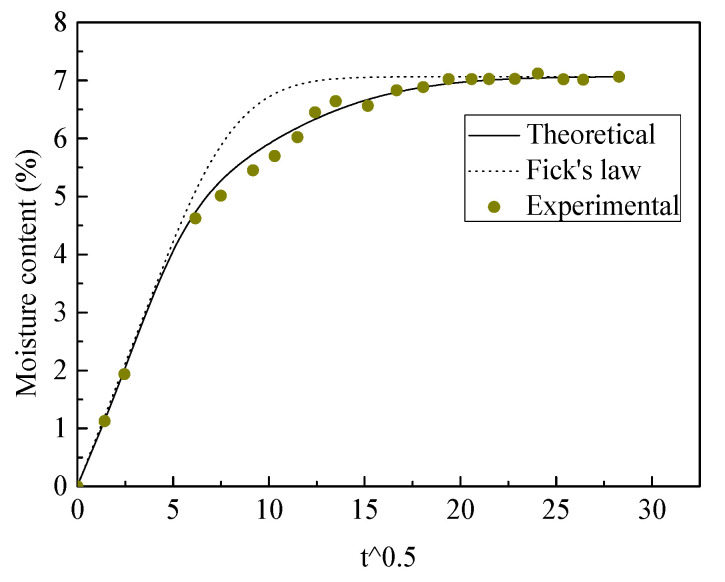
The experimental and theoretical moisture diffusion results for composite b.

**Table 1 polymers-13-02347-t001:** The geometry parameters of the specimens.

Dimension	*a* (mm)	*b* (mm)	*c* (mm)	*d* (mm)
Plain a	0.852	1.514	1.926	1.926
Plain b	0.852	1.514	2.628	2.628

**Table 2 polymers-13-02347-t002:** The dimension parameters of the three cases.

Case	*a*_1_–*a*_2_	*a*_2_–*a*_3_	*a* _3_
1	0.2	1.4	0.4
2	0.6	0.8	0.6
3	1	0.2	0.8

**Table 3 polymers-13-02347-t003:** The geometry parameters of the three cases for the unit cell.

Case	*a*	*b*	*c*
1	1	3	2
2	1	3	1
3	0.5	3	2

**Table 4 polymers-13-02347-t004:** The moisture diffusion parameters of the specimens.

Parameters	$D_{1}({\mathrm{mm}}^{2}/h)$	$D_{2}({\mathrm{mm}}^{2}/h)$	$D_{r}({\mathrm{mm}}^{2}/h)$	$M_{f}(\%)$	$M_{r}(\%)$
Composite	0.0216	0.0165	0.0045	8.6	1.4

## Data Availability

Not Applicable.

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
