# Peer review of "Predictions of Moisture Diffusion Behavior of Cellulose-Fiber-Reinforced Plain Weave Epoxy Composites"

_polymers, 2021, doi:10.3390/polym13142347_

Round 1

Reviewer 1 Report

This manuscript report the moisture diffusion behavior of cellulose fiber reinforced plain weave composites through finite element analysis and experiments. The results show that the predictions can effectively characterize the moisture diffusion behavior of cellulose fiber reinforced plain weave composites. I have some suggestions:

  • The topic of this manuscript is suggested to be “Cellulose fiber plain weave reinforced epoxy composites” .
  • P9 “this boundary condition was discussed in our last article [22]”, however, this reference do not correspond to the content described, especial the author name.
  • The unit of the moisture uptake is missed in Fig.6-14

Author Response

Response to Reviewer 1 Comments

Point 1: The topic of this manuscript is suggested to be “Cellulose fiber plain weave reinforced epoxy composites”.

Response 1: Thank you for the comment. Now, we have revised the topic as “The predictions on moisture diffusion behavior of cellulose fiber reinforced plain weave epoxy composites”.

Point 2: P9 “this boundary condition was discussed in our last article [22]”, however, this reference do not correspond to the content described, especial the author name.

Response 2: Thank you for the comment. Taking the advice, we have revised this reference to [28], the detail is shown as “this boundary condition was discussed in our last article [28]”

Point 3: The unit of the moisture uptake is missed in Fig.6-14

Response 3: Thank you for the comment. According to the suggestion, we have added the unit “%” for the moisture uptake in Fig.6-14.

Reviewer 2 Report

The authors evaluate the moisture diffusion behavior of cellulose fiber reinforced plain weave composite is predicted by a novel superposition method considering the micro weave pattern. The whole moisture uptake of the composite is treated as a moisture absorption superposition of fiber bundles part, resin part, undulated fiber bundles and resin rich part in the unit cell.

This is an interesting study and this Reviewer has some reservations with regards to both readability and technical aspects, as follows:

  1. The English needs revision throughout the manuscript. Several sentences are a bit difficult to understand. Some words need replacement, such as forecasted with predicted;
  2. The authors begin the Introduction with “Since cellulose fiber  reinforced  composites  provide  good  mechanical  properties”. This is definitely not true and must be amended. Any cellulosic fiber has lower mechanical properties than any synthetic fiber, even glass. Please check for some nice ways to introduce it in “Study of hybrid intralaminate curaua/glass composites”
  3. Still in the Introduction, please mention the natural fibers used in composites along with well-established papers, such as curaua (Hybridization effect on the mechanical properties of curaua/glass fiber composites); sisal (Mechanical behavior and correlation between dynamic fragility and dynamic mechanical properties of curaua fiber composites); jute (Jute fiber reinforced epoxy composites and comparison with the glass and neat epoxy composites).
  4. State-of-the-art on moisture diffusion/absorption: I like the way the authors present it, but discussion on recent studies must be included, such as i) Creep and Residual Properties of Filament-Wound Composite Rings under Radial Compression in Harsh Environments; ii) Carbon fiber-reinforced epoxy filament-wound composite laminates exposed to hygrothermal conditioning
  5. Unit cell: how are the voids taken into account in your model?
  6. The waviness presented in Fig 2 looks too coarse. Should it not be smoother?
  7. Fig 5 should be removed as it is too basic and does not bring any useful info to the understanding of your methodology;
  8. Results: excellent. I just wonder how they would look like in case you want to simulate moisture of carbon/epoxy composites, such as the suggested paper (i) in comment 4.

Overall, this is an excellent work.

Recommendation: Major revision.

Author Response

Response to Reviewer 2 Comments

Point 1: The English needs revision throughout the manuscript. Several sentences are a bit difficult to understand. Some words need replacement, such as forecasted with predicted;

Response 1: Thank you for the comment. We have polished the English writing of the whole manuscript carefully, including problems related to understanding of the sentences, the use of similar words, typographical errors, and tense etc.

Point 2: The authors begin the Introduction with “Since cellulose fiber reinforced composites provide good mechanical properties”. This is definitely not true and must be amended. Any cellulosic fiber has lower mechanical properties than any synthetic fiber, even glass. Please check for some nice ways to introduce it in “Study of hybrid intralaminate curaua/glass composites”

Response 2: Thank you for the comment. We now have revised this problem according to the suggestion, the revision is given in page 1 line 25:

“Under increasing pressure from environment concern, the market needs more and more recycled materials. Cellulose fibers (such as curaua, sisal, jute) offer the advantages of low relative density, low cost, and good recyclability compared with synthetic fibers [1-3], so their composites are promisingly used in automotive, civil engineering and other fields in the future [4-9].”

Point 3: Still in the Introduction, please mention the natural fibers used in composites along with well-established papers, such as curaua (Hybridization effect on the mechanical properties of curaua/glass fiber composites); sisal (Mechanical behavior and correlation between dynamic fragility and dynamic mechanical properties of curaua fiber composites); jute (Jute fiber reinforced epoxy composites and comparison with the glass and neat epoxy composites).

Response 3: Thank you for the comment. According to the suggestion, we have added these references as shown below in page 1 line 26.

“Cellulose fibers (such as curaua, sisal, jute) offer the advantages of low relative density, low cost, and good recyclability compared with synthetic fibers [1-3],”

Point 4: State-of-the-art on moisture diffusion/absorption: I like the way the authors present it, but discussion on recent studies must be included, such as i) Creep and Residual Properties of Filament-Wound Composite Rings under Radial Compression in Harsh Environments; ii) Carbon fiber-reinforced epoxy filament-wound composite laminates exposed to hygrothermal conditioning.

Response 4: Thank you for the comment. We now have added two references mentioned above, the details are given below:

Page 2 line 48:

“Almeida et al [21] studied the moisture diffusion behavior of carbon fiber reinforced epoxy filament-wound composite laminates exposed to hydrothermal conditioning, the results show that Fick diffusion is observed for these composites.”

Page 2 line 65:

“Eggers et al [24] investigated the moisture diffusion behavior of filament-wound composite rings under harsh environments, the initial experiment results at room temperature and hot water fit Fick’s law, while ±60/±90 composites show a non-Fickian behavior which is fast transient diffusion at the beginning followed by a slower diffusion rate.”

Point 5: Unit cell: how are the voids taken into account in your model?

Response 5: Thank you for the comment. Voids and crack may exist in composites after they were made, the additional moisture uptake caused by voids and crack is a worth considering issue. However, this issue is rather difficult to predict exactly, so we carefully examine the quality of the specimen to ensure that the voids are as few as possible. Thus, the moisture uptake caused by voids in this paper may affect the whole moisture uptake very limited, so we do not consider effect of the voids in our model.

Point 6: The waviness presented in Fig 2 looks too coarse. Should it not be smoother?

Response 6: Thank you for the comment. The waviness of composite can be treated as smoother shape, it is also feasible to treat the waviness as the shape in Fig.2. The simplification of the shape in Figure 2 can make the calculation more convenient, and the simulation results of this simplification of waviness are also validated by some references, such as:

“Influence of applied in-plane strain on transverse thermal conductivity of 0/90 and plain weave ceramic matrix composites”

“Closed-form solutions of the in-plane effective thermal conductivities of woven-fabric composites”

Point 7: Fig 5 should be removed as it is too basic and does not bring any useful info to the understanding of your methodology;

Response 7: Thank you for the comment. Taking the advice, we have removed Fig.5.

Point 8: Results: excellent. I just wonder how they would look like in case you want to simulate moisture of carbon/epoxy composites, such as the suggested paper (i) in comment 4.

Response 8: Thank you for the comment. The carbon/epoxy composite in the suggested paper is filament-wound composite ring. Considering that the length of this composite is far longer than thickness, I think the moisture diffusion in it may be simplified to a 1D diffusion problem, which only considers the moisture diffusion through thickness direction. Then, this 1D diffusion problem is similar to the case described in our paper, but the details should be adjusted since the unit cell of this composite is different from our paper. If the moisture diffusion in length direction is also taken into account, I think that a 3D moisture diffusion model should be used, but this moisture diffusion problem becomes more complicated. Two references of 3D moisture diffusion and moisture diffusivity are listed here:

“Assessment of 3D moisture diffusion parameters on flax/epoxy composites”

“Moisture diffusion and hygrothermal aging in bismaleimide matrix carbon fiber composites—part I: uni-weave composites”

Reviewer 3 Report

The manuscript deals with the moisture diffusion behaviour of cellulose fibre plain weave composites. Specifically a theoretical investigation of this topic is proposed through the development of a superposition method.

The objective of the research is certainly interesting as an appropriate simulation of the water diffusion process can favour the optimization of the design of composite systems reinforced with natural fibres and predict their behaviour in outdoor applications.

However, the language used must be strongly revised to make the research content understandable and attractive even to readers with little or no familiarity with the methodologies adopted. The manuscript is full of long sentences in which, for example, it is advisable to review the use of punctuation. Among these sentences, one is the following:

Page 8, Lines 211-215: "Considering the generality of three phase diffusion equation, the upper face material of part 4 in cellulose fiber reinforced plain weave composite is resin material, its diffusivity is smaller than cellulose fiber tow, however, the diffusivity of upper face resin material of part 4 in synthetic fiber reinforced plain weave composite is larger than fiber tows. "

Furthermore, a careful re-reading of the text will also allow to remove many typewriting errors still present in the text and to rephrase currently tortuous sentences, such as the following:

Page 4, Lines 130-132: "The moisture uptake of 1/2 part 4 is actually situated between minimum equivalence and minimum equivalence, the thickness of upper resin phase is which is illustrated in Figure 3 (e) and this model is named as approximate equivalence. "

Ultimately, while appreciating the scientific potential of this research, major revisions are mandatory to highlight its added value.

Author Response

Response to Reviewer 3 Comments

Point 1: However, the language used must be strongly revised to make the research content understandable and attractive even to readers with little or no familiarity with the methodologies adopted. The manuscript is full of long sentences in which, for example, it is advisable to review the use of punctuation. Among these sentences, one is the following:

Page 8, Lines 211-215: "Considering the generality of three phase diffusion equation, the upper face material of part 4 in cellulose fiber reinforced plain weave composite is resin material, its diffusivity is smaller than cellulose fiber tow, however, the diffusivity of upper face resin material of part 4 in synthetic fiber reinforced plain weave composite is larger than fiber tows. "

Response 1: Thank you for the comment. Taking the advice, we have revised the long sentences problems in the whole paper, for example:

The sentences in page 8 line 219:

“The three phase diffusion equation is solved by Eqs.(16-18), we first utilize FEA to validate the exact solution. Two cases for the moisture uptake of part 4 are considered, one case is D1 (0.005) < D2 (0.02), C1 (0.5) < C2 (1), the other case is D1 (0.02)> D2 (0.005), C1 (1) > C2 (0.5). Two different cases are validated by FEA with different phase thickness, the results are shown in Figure 5 and 6, the dimension parameters are shown in Table 2.”

The sentences in page 4 line 117:

“For the moisture diffusion through thickness direction in part 4, the analysis on the simplified equivalence structure of part 4 is illustrated in Figure 4. We could firstly cut part 4 (Figure 3(a)) in half due to symmetrical property, 1/2 part 4 is depicted in Figure 3(b). Obviously, the moisture absorption of part 4 is twice of 1/2 part 4. From Figure 3(b), it is found that water immersion in upper face of 1/2 part 4 will first diffuse through resin regions, however, water will also diffuse into orange region at the first beginning. This phenomenon will make the water absorption in the initial stage complicated. If orange region represents cellulose fiber tow, the moisture diffusion of 1/2 part 4 will increase slowly at the initial stage, since the diffusivity of cellulose fiber tow is larger than resin phase. Then, the moisture diffusion will quickly rise because of cellulose fiber tow. To predict this moisture diffusion behavior, the fast diffusion of cellulose fiber tow and slow diffusion of resin material will be considered by using a simplified equivalence method. The undulated fiber tow, triangular and trapezoidal resin phase are assumed to be equivalent to rectangle structures, so that the analysis will be simplified as a three phase diffusion problem. The areas of different phases in equivalent model are equal to former fiber tow and resin regions. Next, we will talk about three thickness cases of upper resin:

(1) Minimum equivalence. If we directly ensure that the area of upper triangular resin phase is equal to equivalent model, i.e. the thickness of upper resin in equivalent model is 4/a, the moisture diffusion will be underestimated because moisture diffusion in cellulose fiber tow is not enough considered. We will later prove this phenomenon, this model is named as minimum equivalence as shown in Figure 3(c).

(2) Maximum equivalence. If the thickness of upper resin phase is 0 which is depicted in Figure 3(d), the moisture diffusion will be obviously overestimated. This model is named as maximum equivalence.

(3) Approximate equivalence. As illustrated in Figure 3(e), the moisture uptake of 1/2 part 4 is actually situated between minimum equivalence and minimum equivalence. The thickness of upper resin phase is τa, and this model is named as approximate equivalence.”

Point 2: Furthermore, a careful re-reading of the text will also allow to remove many typewriting errors still present in the text and to rephrase currently tortuous sentences, such as the following:

Page 4, Lines 130-132: "The moisture uptake of 1/2 part 4 is actually situated between minimum equivalence and minimum equivalence, the thickness of upper resin phase is which is illustrated in Figure 3 (e) and this model is named as approximate equivalence. "

Response 2: Thank you for the comment. According to the suggestion, we carefully revised the whole paper, for example:

In page 5 line 141:

“As illustrated in Figure 3(e), the moisture uptake of 1/2 part 4 is actually situated between minimum equivalence and minimum equivalence. The thickness of upper resin phase is τa, and this model is named as approximate equivalence.”

In page 4 line 102:

“The superposition method is depicted in Figure 2. The orange regions are fiber tows, while the empty regions in unit cell are resin materials. This unit cell is divided into 9 parts according to the type of each structure, the details of them are shown in Figure 2(a). Parts 1, 3, 7 and 9 in Figure 2(b) represent tow parts which could be treated as unidirectional composites, and the moisture in them actually diffuses through the transverse direction of unidirectional composites. As shown in Figure 2(c), we could find that parts 2, 4, 6 and 8 have similar structure, i.e., the upper and lower faces in these parts are resin materials while the middle structures in these parts are undulated tows. If c=d, we could find that the moisture absorptions of these parts are equal. The transient moisture diffusion in parts 2, 4, 6 and 8 may be more complicated compared to other parts, due to the undulated tows and different types of materials in them. Thus, the predictions on moisture diffusion of them are key problems in this paper. Part 5 is pure resin material, its moisture uptake obeys Fick’s law, part 5 is shown in Figure 2(d).”

In page 5 line 141:

“As illustrated in Figure 3(e), the moisture uptake of 1/2 part 4 is actually situated between minimum equivalence and minimum equivalence. The thickness of upper resin phase is τa, and this model is named as approximate equivalence.”

Round 2

Reviewer 2 Report

The manuscript can now be accepted for publication.

Reviewer 3 Report

The revised version of the manuscript is acceptable.